# 3D Printing of Macro Porous Sol-Gel Derived Bioactive Glass Scaffolds and Assessment of Biological Response

**DOI:** 10.3390/ma14205946

**Published:** 2021-10-10

**Authors:** Ricardo Bento, Anuraag Gaddam, Párástu Oskoei, Helena Oliveira, José M. F. Ferreira

**Affiliations:** 1CICECO―Aveiro Institute of Materials, Department of Materials and Ceramic Engineering, University of Aveiro, Santiago University Campus, 3810-193 Aveiro, Portugal; ricardobento@ua.pt (R.B.); anuraagg@ua.pt (A.G.); 2Instituto de Física de São Carlos, Universidade de São Paulo, São Carlos 13566-590, SP, Brazil; 3Department of Biology & CESAM, University of Aveiro, Santiago University Campus, 3810-193 Aveiro, Portugal; parastu.oskoei@ua.pt (P.O.); holiveira@ua.pt (H.O.)

**Keywords:** bioactive glasses, alkali-free, sol-gel, bone regeneration, tissue engineering

## Abstract

3D printing emerged as a potential game-changer in the field of biomedical engineering. Robocasting in particular has shown excellent capability to produce custom-sized porous scaffolds from pastes with suitable viscoelastic properties. The materials and respective processing methods developed so far still need further improvements in order to obtain completely satisfactory scaffolds capable of providing both the biological and mechanical properties required for successful and comprehensive bone tissue regeneration. This work reports on the sol-gel synthesis of an alkali-free bioactive glass and on its characterization and processing ability towards the fabrication of porous scaffolds by robocasting. A two-fold increase in milling efficiency was achieved by suitably adjusting the milling procedures. The heat treatment temperature exerted a profound effect on the surface area of mesoporous powders. Robocasting inks containing 35 vol.% solids were prepared, and their flow properties were characterized by rheological tests. A script capable of preparing customizable CAD scaffold geometries was developed. The printing process was adjusted to increase the technique’s resolution. The mechanical properties of the scaffolds were assessed through compressive strength tests. The biomineralization ability and the biological performance were assessed by immersing the samples in simulated body fluid (SBF) and through MTT assays, respectively. The overall results demonstrated that scaffolds with macro porous features suitable for bone ingrowth (pore sizes of ~340 μm after sintering, and a porosity fraction of ~70%) in non-load-bearing applications could be successfully fabricated by 3D printing from the bioactive glass inks. Moreover, the scaffolds exhibited good biomineralization activity and good biocompatibility with human keratinocytes, suggesting they are safe and thus suited for the intended biomedical applications.

## 1. Introduction

For years, subtractive manufacturing, consisting of removing, treating, and shaping a feed material into a product with desired properties and shapes, has been the paradigm in manufacturing and industrial processes. However, in the recent past, technological developments in the manufacturing and digital fields have led to the candidacy of additive manufacturing (AM) as a potential gateway into a new industrial revolution [1]. AM is now, more than ever, the target of numerous research efforts in multiple fields to assess its potential, limitations, and feasibility [2,3]. Its impact is not merely on the industrial sector [2,3,4,5,6,7] but also economics [4,8,9,10,11], fiscal policy, and social welfare [9] as the manufacturing capability of various products suddenly becomes possible in the household. Perhaps the biggest impact of AM is its ability to create intricate and easily customizable geometries not feasible through traditional methods, which is particularly useful when coupled with finite element method (FEM) analysis to produce more efficient parts, to model unique designs for expositions, or to produce extremely precise implants that can be custom-fitted into patients, opening up a whole new paradigm in biomedical engineering, although the regulation and success rate of these custom devices presents a possible legal nightmare [1].

Robocasting is a 3D printing technique that uses a triaxial (XYZ) dispenser to extrude a polymeric, metallic, or ceramic slurry in a layer-by-layer fashion, and the extruded filament fuses with the existing layers. This simple technique relies mostly on the viscoelastic properties of the pastes, which impose design limitations since arching or overhanging features as well as hollow segments are impossible to reproduce without the use of supporting or sacrificial materials. This simplicity allows, however, great flexibility in the choice of printing materials [12]. Ceramic slurries, also called inks, are traditionally composed of ceramic powders—typically with particle size distributions and average sizes ranging from submicrometric to several microns of diameter—dispersed in a liquid (such as water) with the aid of minimal amounts of selected dispersants, binders, and coagulating agents [13]. The reliance on the viscoelastic properties of the printing inks puts the rheology and viscosity of the paste in a central position. Robocasting pastes should be homogeneous so as to avoid any change of pressure in the syringe and uneven densities in the green body. Cesarano et al. [14] established three criteria that a ceramic ink should follow: (i) the viscosity should allow it to flow through a small orifice when submitted to modest shear rates; (ii) the ink must stop flowing after being deposited onto the substrate and, therefore, exhibit a fast recovery of the internal structure; (iii) the deposited filaments must be capable of supporting the upper layers without yielding [13]. For this, the ideal paste should be shear-thinning (pseudoplastic) when flowing and shear-thickening (dilatant) upon deposition. 

The printing pastes should have high solids loadings to prevent the formation of cracks upon the drying and sintering processes [15]. To achieve such demanding flow properties, different approaches have been employed. Depending on the specifications of the solid and liquid components involved in the paste formulation, namely on the solid–liquid interactions and the packing ability of the powder, the starting suspensions can be prepared with a high solids loading (over 50 vol.%) in water with a very minute quantity of processing additives. The common approach is searching for a trade-off between a high solids loading and a relatively low viscosity—usually the use of dispersant concentrations just below the pseudo-plastic to dilatant transition—to enable the achievement of a high degree of homogeneity. However, such suspensions are not suitable for printing as the individual particles are prone to undergo size segregation under shear, which may clog the extrusion nozzles. The dispersing agents alone can hardly enable us to adjust and control the rheological properties of suitable printing pastes. Accordingly, other processing additives need to be added in order to cause drastic changes on the rheological properties, transforming a fluid suspension into an extrudable paste. With this purpose, thickening agents are usually employed to increase the viscosity of the dispersing liquid to hinder particle segregation under shear. They also act as binders, enhancing the mechanical properties of the green parts [14]. Furthermore, the addition of a coagulating agent having an anionic character opposite to that of the dispersant is commonly required to increase the stiffness of the ink and to confer it the above referred viscoelastic properties required for printing [15]. Therefore, the required rheological characteristics for printing rely on the overall complex interactions taking place within the pasty-like system, which also affect the following processing steps, such as drying and sintering [15]. Different strategies were attempted to improve the printability of different materials by robocasting. High solids loading pastes were developed using anionic polyelectrolytes as dispersants to electro-sterically stabilize the starting suspensions and provide good homogenization, cellulose-based binders, and a cationic polyethylenimine (PEI) as a coagulation agent [15,16,17,18,19]. Recently, the use of hydrogels, most notably Pluronic F127, has been given attention in the processing of difficult-to-disperse powders as an alternative approach to obtaining the highly printable pastes prepared thereof since the interactions among the imbuing particles lose relevance and, therefore, any powdered material is theoretically mixable [20,21]. Besides, Pluronic F127-based pastes are highly stable and can be stored over long durations, which, coupled with their ease of preparation, could facilitate the production of such pastes for commercial purposes [12,22]. Previous studies on the biological properties assessment of sol-gel derived bioactive glasses are scarce. A four-component, high-silica sol-gel glass with a molar composition of 67SiO_2_–24CaO–2.5Na_2_O–2P_2_O_5_ containing small amounts of CuO and La_2_O_3_ revealed that co-doping with Cu^2+^ and La^3+^ could benefit the viability of C13895 lymphoblast cells [23]. So far, no biocompatibility studies have been carried out and reported for the better-balanced bioactive glass composition disclosed here, justifying the pertinence of evaluating its biological performance.

## 2. Materials and Methods

### 2.1. Bioactive Glass Synthesis

A quaternary bioactive glass with composition 60SiO_2_–34CaO–4MgO–2P_2_O_5_ (mol %) was synthesized by the sol-gel technique from the following reagents: tetraethylorthosilicate (TEOS, Si(O_2_H_5_), ≥98%) supplied by Sigma-Aldrich)(Sintra, Portugal), trietylphosphate (TEP, O_4_P(C_2_H_5_O), ≥98%) supplied by MERCK-Schuchardt (Darmstadt, Germany), magnesium nitrate hexahydrate [Mg(NO_3_)_2_·6 H_2_O) supplied by Scharlau (Barcelona, Spain)], and calcium nitrate tetrahydrate [Ca(NO_3_)_2_·4 H_2_O) supplied by Panreac (Barcelona, Spain)] as precursors for Si, P, Mg, and Ca oxides, respectively. Nitric acid (HNO_3_ ≥ 65%) supplied by Labkem (Dublin, Ireland) was used for catalysts to promote the hydrolysis of network Si and P precursors. The preparation of the batches was planned to yield 0.2 mol of bioactive glass by mixing 1.46 g of TEP, 25.00 g of TEOS, 2.05 g of magnesium nitrate, and 16.06 g of calcium nitrate. Two separate aqueous solutions were initially prepared by adapting a procedure reported by Ben Arfa et al. [24]. The solution of the network modifiers was prepared by adding the required amounts of Ca(NO_3_)_2_·4 H_2_O and Mg(NO_3_)_2_·6 H_2_O to 20 mL of deionized water under magnetic stirring for 1 h. The network precursors were also dissolved in 20 mL of deionized water acidified with two drops of concentrated nitric acid under magnetic stirring for 1 h. Then, both solutions were mixed and magnetically stirred for further 1 h before pouring into Petri dishes, and stored in an oven for 24 h at 100 °C to promote a relatively rapid sol-gel transition. As obtained, the transparent gel was quickly crushed in small granulates with a spatula and then put again in the same oven for drying. The xerogel was first ground into a fine powder using an agate pestle and mortar and then calcined at 600 °C. The structural features of powders of this bioactive glass calcined at different temperatures (600 °C, 700 °C, and 800 °C) have already been disclosed elsewhere [25], and they reveal that it remains completely amorphous within this entire calcination and sintering temperature range (600–800 °C). The biomineralization ability of the same powders immersed in simulated body fluid (SBF) was also reported in that manuscript. The readers are requested to consult this article for the related information. 

### 2.2. Milling Procedures

Milling procedures were adapted from the guidelines provided by Ben Arfa et al. [26,27] using a planetary mill (Ceramic instruments, type S2-1000, Sassuolo-Italy), at 390 RPM. Ethanol or acetone were tested as liquids for wet milling at the bioactive glass-to-liquid weight ratio of 1:1 in a sintered alumina jar of 300 cm^3^ capacity using yttria-stabilized zirconia balls with a 10 mm diameter (Tosoh, Tokyo, Japan). A mass of 15 g of bioactive glass was milled at a time (Table 1). 

### 2.3. Particle Characterization

The particle size and its distribution were measured by laser diffraction using a particle size analyser (Coulter LS particle size analyser, Beckman Coulter, Mississauga, ON, Canada) with triplicate measurements. Average powder density was determined using triplicate measurements performed with a helium AccuPyc 1330 Pycnometer (Micromeritics Instrument Corporation, Norcross, GA, USA). 

### 2.4. Paste Formulations

Hydrogels containing 33 wt.% Pluronic F-127 in deionized water were chosen as processing media for the preparation of the robocasting inks. Two different volumes were prepared at a time: 5 mL and 10 mL stock solutions. Zirconia balls were added to both solutions to aid in the mixing process: 3 balls with a 5 mm diameter to the 5 mL stock, 6 balls with 5 mm, and an additional 3 balls with 10 mm diameter to the 10 mL stock solution. Pastes were produced by adding the powders to the Pluronic F-127 hydrogels and mixing in a planetary centrifugal mixer (ARE-250, Thinky Corp., Tokyo, Japan) at different RPM, resulting in homogenized inks.

### 2.5. Rheological Characterization of the Inks

Rheological measurements of the inks were taken using a Kinexus Pro+ Rheometer (Malvern Instruments, Westborough, MA, USA). A cone-plate geometry (4°/40 mm) with a gap of 150 μm was adopted to test their flow behaviours under the rotational mode. Their viscoelastic properties were assessed using parallel plates with a gap of 1 mm under oscillatory mode at the frequency of 1 Hz. 

### 2.6. Printing Parameters

A robocasting printer (3-D Inks, Stillwater, Littleton, CO, USA) on a 4-axis setup (XYZU) was used to print the scaffolds. The piston mechanism was custom built. The paste was loaded onto syringes (Luer locker, 3 mL, Nordson, Westlake, OH, USA) coupled with extruding nozzles (Optimum general purpose dispense-tips, internal diameter 410 µm, Nordson, Westlake, OH, USA). Printing atmosphere was controlled using an air humidifier (SHF 911GR, Sencor, Ricany Czech Republic) and air conditioning, resulting in relatively constant printing temperatures of 25 °C and an average humidity of 80%. Printed specimens were layered on a 100 × 100 × 1 mm alumina plaque with a thin layer of anti-sticking agent (Zhengzhou Keija Furnace CO., LTD, Zhengzhou, China) at a printing speed of 10 mm/s.

Proprietary RoboCAD software (3-D Inks, Stillwater, OK, USA) was used in freeform CAD modelling to produce square scaffolds. Specimens were left for 1 day at 38 °C to allow even and slow drying before being stored at 100 °C in an oven.

### 2.7. Debinding, Sintering, and Morphological Observation of the Scaffolds

Thermogravimetric analysis (TGA) was employed to determine the parameters for debinding and sintering using a Netzsch STA 449F1 calorimeter. Dilatometry measurements (Bahr Thermo Analyzer DIL 801L, Hillhorst, Germany) under a heating rate of 10 °C min^−1^ were carried out using rectangular samples with 11 mm × 4 mm × 2 mm prepared by robocasting. Thereafter, the bioactive glass scaffolds were debinded at 400 °C for 4 h with a heating rate of 0.5 °C min^−1^, followed by sintering at 600, 700, 800, and 900 °C for 4 h with a heating rate of 0.5 °C min^−1^. For the morphological observation, the scaffolds were covered with a thin (15 nm) carbon layer using a thin film deposition system (PVD 75, Kurt J. Lesker Co., Jefferson Hills, PA, USA) before being examined using a Hitachi S4100 Scanning Electron Microscope with a 15.0 kV accelerating voltage.

### 2.8. Assessment of the Mechanical Properties through Compressive Strength Tests

The compressive strength of the scaffolds was determined under uniaxial testing with 5 samples using a universal testing machine (AG-IS10kN, Shimadzu, Kyoto, Japan) at a constant speed of 0.5 mm/min in the perpendicular direction to the printing plane with a 5 kN cell.

### 2.9. Assessment of the Biological Properties through In Vitro Cytotoxicity Assays

The cytotoxicity of the bioactive glass scaffolds was assessed using the human osteoblast-like cell line MG-63, kindly provided by INEB, University of Porto (Porto, Portugal. Cells were cultured in Dulbecco’s modified Eagle’s Medium (DMEM), supplemented with 10% fetal bovine serum (FBS), 2 mM L-glutamine, 1 × 10^4^ U/mL penicillin/streptomycin, and 250 μ g/mL fungizone (all medium components from Life Technologies, Carlsbad, CA, USA) at 37 °C in a 5% CO_2_ humidified atmosphere. Extracts from the different scaffolds were prepared by placing a scaffold in 10 mL DMEM culture medium, followed by incubation at 37 °C in a 5% CO_2_ humidified atmosphere for 24 h. Cytotoxicity of the extracts was assessed by the colorimetric MTT assay [28]. Cells were seeded in a 96-well plate at 1 × 10^3^ cells per well, and, after cell adhesion, the culture medium was replaced by the different scaffold extracts at 100% concentration or diluted to 50% with fresh DMEM medium. MG-63 cells exposed to DMEM medium were used as negative control. Cells were further incubated for 24 h at the same temperature and atmosphere conditions as before. After incubation, 50 µL of an MTT (Sigma-Aldrich, St. Louis, MO, USA) solution (1 mg/mL in PBS pH 7.2), a yellow tetrazolium dye, were added to each well followed by further incubation for 4 h. Living cells reduce the MTT to purple formazan crystals. Then, the culture medium with MTT was removed and replaced by 150 µL of dimethyl sulfoxide (DMSO) and placed in a shaker for 2 h to dissolve the formazan crystals. Absorbance measurements were used to calculate cell viability according to Equation:$$Cell\;Viability\left(\%\right)=\left(\frac{AbsSample-AbsDMSO}{AbsControl-AbsDMSO}\right)\times 100$$

The absorbance of the samples was measured with a BioTeK Synergy HT plate reader (Synergy HT Multi-Mode, BioTeK, Winooski, VT, USA) at 570 nm with blank corrections.

## 3. Results and Discussion

### 3.1. Sol-Gel Synthesis 

Homogeneous and transparent solutions were obtained from all the precursors using deionized water as a single solvent, as described in our work [25]. The dissolution of the Si and P network precursors was facilitated by the addition of two drops of concentrated nitric acid (HNO_3_ ≥ 65%), which also acted as catalysts to promote their hydrolysis. The solution of the network modifiers soon became transparent due to their high solubility. The solution of the network formers became transparent in less than 20 min of magnetic stirring. The mixing of both separately prepared solutions resulted in a homogeneous transparent solution, which was kept under magnetic stirring for another hour. After being poured into Petri dishes and moved to an oven at 100 °C, this overall solution underwent a relatively rapid sol-gel transition, which is useful to prevent the chemical segregation of the components, generating a transparent gel. The preliminary crushing of the fresh gel proved to be very helpful, facilitating the subsequent further grinding of the xerogel.

### 3.2. Milling Procedures

The milling of the bioactive glass powder that was calcined at 600 °C was performed by first crushing it with an agate pestle and mortar until the particles could pass through a 150 μm sieve. These particles were then further milled in ethanol with a weight ratio of 1:1 using 10 mm diameter zirconia balls at a fixed weight ratio of 1:10, as described above and elsewhere [26,27]. After 1 h of milling, the effective removal of all the liquid and powder from the jar to a Petri dish for the subsequent drying step was difficult. To overcome this inconvenience, acetone was tested as an alternative milling liquid at the same 1:1 weight ratio. After 1 h of milling in acetone, the powder could be easily removed from the jar and immediately brushed against a 63 μm sieve, circumventing the need for the drying step. This procedure was simpler, more time-effective, and presented higher efficiency and material recovery, as seen in Table 1. The average density of the milled bioactive glass powder is reported in Table 2. 

The particle size distribution of the milled powder determined by laser scattering is present in Figure 1. A relatively broad distribution within the range of 0.2 to 20 μm can be observed with a slight right skewness and a mean particle size of 4.63 ± 3.58 μm.

### 3.3. Paste Preparation

Pluronic F-127 hydrogels provide a more rapid and simpler way of producing pastes for robocasting, circumventing the need to test out multiple concentrations of processing additives. On the other hand, their relatively high viscosities limit the maximum achievable solids loading, which harms the capacity to produce dense, monolithic pieces. However, as this study focuses on the production of porous scaffolds for tissue regeneration, this compromise was taken. In this work, six different solids loadings were initially attempted: 10, 20, 25, 30, 35, and 40 vol.% to understand their effects on the flow properties of the pastes. At 40 vol.%, the viscosity was too high, hindering the preparation of a properly mixed and homogeneous ink. 

The pastes were mixed by first storing the Pluronic F-127 hydrogel in a freezer until it became more fluid. Then, all the powder was added and mixed at once at high rotations (1400–1600 RPM) for 2 min, resulting in homogenous pastes.

### 3.4. Rheological Properties of the Inks

The printing ability of the pastes was assessed through rheological tests. Figure 2 displays the apparent viscosity versus shear rate curves of the pastes containing solids loadings varying from 10 to 35 vol.%. An expected gradual increase in the viscosity with increasing additions of solid particles is observed within the shear rate range from about 0.3 to 100 s^−^^1^, especially within the interval from 0.3 to 10 s^−^^1^. Furthermore, all the plotted curves exhibit almost linear variations of viscosity with increasing shear rates. This shear-thinning behavior is an important requisite for 3D printing. The curves tend to overlap under near zero shear viscosity conditions (shear rates < 0.3 s^−^^1^), reflecting the intrinsic interactions between the hydrophobic and hydrophilic segments of the Pluronic F-127 chains in the hydrogel under rest conditions. For the deformation to start, the polymeric network needs to undergo a certain structural remodelling to enable the polymeric chains to gradually align along the flow direction, which is likely to require an extra applied stress (yield stress). This is reflected in the slight increase in the apparent viscosity observed for very low shear rates before it drops due to shear-thinning. Moreover, the flow curves of the pastes containing 20, 25, and 30 vol.% solids tend to almost overlap within the interval from about 10 to 100 s^−^^1^, meaning that their rheological properties under these conditions are mostly dictated by the intrinsic interactions between the hydrophobic and hydrophilic segments of the Pluronic F-127 chains. However, with a further increase in the solids loading to 35 vol.%, the system becomes more crowded with bioactive glass particles, which disturb the polymeric chain interactions while contributing more decisively to the stiffening of the paste, explaining the jump observed in the respective curve in Figure 2.

The pastes were also tested in oscillatory mode. The lower solids loadings (<30 vol.%) did not confer sufficient stiffnesses to the inks to enable good shape retention capability of the extruded filaments, as can be deduced from the corresponding viscoelastic parameters of the inks plotted in Figure 3. The elastic modulus (G′) significantly increases with increasing solid concentrations, reaching a plateau at about 10 kPa at 20–25 vol.% and at about 30–40 kPa at 30–35 vol.%, which are below the satisfactory level. For good printing and shape retention capability, G′ values should be at least one order of magnitude higher than those that are usually required [29].

The different possible strategies for ink preparation must be carefully selected considering the surface chemistry of the materials to be processed [20,21,30,31]. For example, the standard approach, using an anionic dispersant and a cationic coagulant, such as PEI, proved to be completely useless for printing 45S5 bioactive glass scaffolds [30,31]. This problem could be overcome by using long chain length (250,000 g mol^−^^1^), sodium carboxymethyl cellulose (CMC) as a single processing additive (dispersant, binder, and stiffening agent). The use of Pluronic F-127 hydrogels as a dispersion medium also offers this obvious advantage of simplifying the ink formulation from the processing additives viewpoint.

This option for Pluronic F-127 hydrogels is fully justified when the powder to be dispersed undergoes some ionic leaching that would promote early and undesirable coagulation phenomena and unstable inks [21], or when dealing with poor wetting powders, such as metallic Al [20]. On the other hand, when the powder undergoes hydrolysis and non-stoichiometric dissolution reactions, the standard approach of using dispersant and PEI can only be used for surface-treated powders against hydrolysis [32]. 

Moreover, the option for Pluronic F-127 hydrogels also brings some associated drawbacks, including an increased difficulty in preparing inks with high solids loadings. In the present case, the most concentrated ink (40 vol.% solids) was difficult to prepare with a high homogeneity. The intrinsic porosity of the bioactive glass particles retaining some dispersing liquid is likely to account for this [33]. The rheological properties of inks based on Pluronic F-127 hydrogels being less sensitive to the overall solids volume fraction and other processing parameters make the processing conditions trickier to tune and more difficult, and some further refinements will be required in future studies. 

### 3.5. Printing 

The CAD models of scaffolds were built using RoboCAD software. A model consisting of a squared lattice with 10 mm × 10 mm × 4 mm dimensions with a meandering path inside it and each rod being spaced by a length equal to its diameter was designed, as seen in Figure 4. A small, solid rectangular piece was also freeform modelled for use in dilatometry tests, as seen in Figure 5. Based on this model, the porosity of the scaffold was estimated according to this CAD model. The mathematical model developed for deriving the porosity fraction is presented in the Appendix A.

The printing was carried out by loading the pastes into syringes with a couple nozzle of 410 µm inner diameter. Air conditioning and a humidifier were used to keep the printing environment at approximately 25 °C and 80% relative humidity. The step in the Z direction that takes place when the printer is switching to the next layer was initially set in the CAD model at ΔZ = 0.322 mm. However, this degree of overlap was revealed to be excessive because the inks are not so stiff, as shown in Figure 3, and the deposited filaments would undergo some flowing while their internal structure is being recovered. Therefore, this parameter was changed from the default value of 0.322 mm to 0.410 mm in order to prevent the dragging of the previously deposited underlying layers and the geometrical deformation of the printing parts. This also means that the real overlap among the different deposited layers can exclusively be attributed to the incipient flowing undergone by the deposited filaments while their internal structure is being recovered. The benefits of this change in the set overlap degree from ΔZ = 0.322 mm to ΔZ = 0.410 mm in the model can be observed in Figure 4. The parts on the left-hand side printed with ΔZ = 0.322 mm present high geometrical deformations in contrast with those on the right-hand side printed with ΔZ = 0.410 mm. According to the CAD and mathematical models, these scaffolds have an internal porosity of ~70%. Although this value is specific to a green body, as long as the shrinkage is isotropic during sintering, the sintered scaffolds should also have the same internal porosity. In the current case, it is reasonable to presume isotropic shrinkage, at least within the amenable limits of the mechanical stresses that would otherwise develop during drying and sintering, leading to the bending and cracking of the scaffolds, putting their integrity in danger. Since bending and cracking were not observed, and isotropic shrinkage can be assumed. Thus, the porosity calculation for the green body should also hold for the sintered body. 

It can be clearly seen that a significant improvement of geometrical definition was achieved for the bioglass scaffold and for the specimen for dilatometry measurements printed with ΔZ = 0.410 mm, i.e., when a null overlap was set in the CAD model, allowing the real overlap to be dictated just by the incipient flowing of the newly deposited filament. Therefore, this seems to be an interesting expedient to help remediating the unsatisfactory stiffness of robocasting inks based on Pluronic F-127 hydrogels. However, refinements in the ink preparation process, such as gradually adding the powder in different steps intermediated with mixing steps at suitable high rotations and for longer time periods, could be explored in future studies aiming at homogeneous inks with higher solids loadings and degrees of stiffness. 

### 3.6. Thermal Treatments

Scaffold debinding was first attempted at 300 °C for 2 h, resulting in a grey tone scaffold, likely due to carbon residues from the incomplete burn-out of the Pluronic F-127. Therefore, to select the debinding and sintering parameters, a previously dried (at 100 °C) and powdered paste was submitted to thermal analysis within the temperature range from 25 to 900 °C, performed at a heating rate of 10 °C min^−1^. The curves of the thermogravimetric analysis displayed in Figure 6 exhibit a broad thermal band ranging from 150 to 600 °C with DTG peaks at ~200 and ~320 °C and ~400 °C, which correspond to the faster degradation rates of the polymer in the TGA curve. The small weight loss up to ~200 °C and peaking at this temperature is attributed to the desorption of water [34]. The onset of polymer thermal decomposition occurs at about ~320 °C, as shown in the TDA curve relative to the pure Pluronic F-127 sample. This thermal behaviour agrees with the observations made by other authors [34,35]. The peak at 700 °C is likely due to the burn-out of residual carbon. The analysis suggests that a complete burn-out of the Pluronic F-127 is likely to occur at 400 °C under slow heating, in accordance with the literature [36]. Heating the printed scaffolds at a rate of 0.5 °C min^−1^ up to 400 °C and plateauing for 4 h produced light-coloured scaffolds, indicating an apparent complete burn-off. 

Before sintering, a dilatometric test was carried out using a rectangular specimen with ~10.8 mm of length (shown in Figure 5). The specimen was previously debinded at 400 °C so it could be placed into the dilatometric analyser without fracture. The dilatometry curve displayed in Figure 7 shows that slow dimensional changes only started to occur at about 450 °C and gradually accelerated up to 760 °C, followed by an accentuated shrinkage up to 850 °C, and reached the minimal dimensions at approximately 900°C. Moreover, the preliminary dimensional assessments of the printed scaffolds before and after sintering revealed that anisotropic shrinkage was not an issue here. Therefore, the sintered scaffolds also have the same internal porosity as the green body, ~70%.

### 3.7. Preliminary Mechanical Properties Assessment through Compressive Strength Tests

Compressive strength tests were performed on five representative scaffolds. According to the followed ISO 17162 (2014), the specimens must have a height larger than the width and length values, which is not the case. However, the main purpose of these tests was evaluating the reproducibility of the mechanical properties and to infer if they could be suitable for the intended applications in bone regeneration and tissue engineering. The bioglass scaffolds sintered at 600 and 700 °C were fragile and likely to break when handled, so they were not selected for the compressive tests. Their poor mechanical strength can be attributed to the insufficient densification degrees of the struts, as deduced from Figure 7. The values for the scaffolds sintered at 800 °C are present in Table 3. It can be seen that the average compressive strength (8.39 ± 0.98 MPa) is well within the range reported for human cancellous (trabecular) bone (2–12 MPa) [37,38].

### 3.8. Biocompatibility Assays 

To assess the biocompatibility of the different scaffolds and their suitability for biological application, MG-63 cells were exposed to culture medium extract where the scaffolds were incubated for 24 h. Viability values for the four different samples can be seen in Figure 8. For all the scaffolds, the cell viability values are above the 70% threshold, which, according to ISO 10993-5:2009(E) (2009), suggests they are safe and thus suitable for biomedical applications. In the cells exposed to 100% extract, the cell viability varies from 80.9% in samples sintered at 600 °C to 90.5% in samples sintered at 900 °C. Surprisingly, the samples sintered at 900 °C had the best results compared to those that were sintered at 600 °C, displaying the lowest cell viability. However, the scaffolds sintered at 700 °C had slightly higher cell viability values than those sintered at 800 °C. Therefore, a trend cannot be established between sintering temperature and cell viability. Moreover, the observed changes in the trend are lower than the error bars, which further makes the trend uncertain. These data highlight the need for a thorough characterization of the sintered scaffolds in future studies.

## 4. Conclusions

Porous bioactive glass scaffolds with macro pore size, macro pore volume fraction, and mechanical properties suitable for bone ingrowth in bone regeneration and tissue engineering applications were successfully fabricated by robocasting. The amorphous sol-gel derived bioactive glass powder with the composition of SiO_2_-CaO-MgO-P_2_O_5_ was calcined at 600 °C, milled down to a mean particle size of 4.63 ± 3.58 µm, and dispersed in a 33 wt.% concentrated Pluronic F-127 hydrogel to form printing inks with various solids loadings. The assessment of the rheological properties of the inks revealed the following salient features: (i) the inks tend to exhibit shear-thinning behaviours that are required for 3D printing; (ii) the high viscosity of the background Pluronic F-127 hydrogel limits the quantity of solids that can be dispersed, hindering the achievement of homogeneous inks with high solid volume fractions and degrees of stiffness; (iii) the maximum solids loading for the printable inks was limited to 35 vol.%; (iv) the elastic modulus of the ink containing 35 vol.% of the bioactive glass powder was at least one order of magnitude lower than the values usually required for granting good shape retention capability to the extruded filaments. Therefore, the depositing filaments still undergo some incipient flowing while their internal structure is being recovered, allowing a sufficient overlap with the previously printed layer. This overlap degree, together with that imposed by the CAD model, was revealed to be excessive and could lead to printed parts with geometrical deformations. This problem could be overcome by setting the overlap degree in the CAD model at zero. 

The debinding of the printed parts needs to be conducted under low heating rates to properly degrade the polymeric chains while keeping the integrity of the scaffolds. A sintering temperature of at least 800 °C is necessary for conferring suitable compressive strength values to the porous scaffolds, matching those of human cancellous (trabecular) bones. The ability of the material to form a hydroxyapatite layer after only 7 days of SBF treatment was an attractive feature already revealed in the previous study. The biological assessment revealed cell viability values over the 70% threshold and up to ~90%, which strongly supports that these bioactive glass scaffolds are safe and suitable for biomedical applications, bone regeneration, and tissue engineering.

## Figures and Tables

**Figure 1 materials-14-05946-f001:**
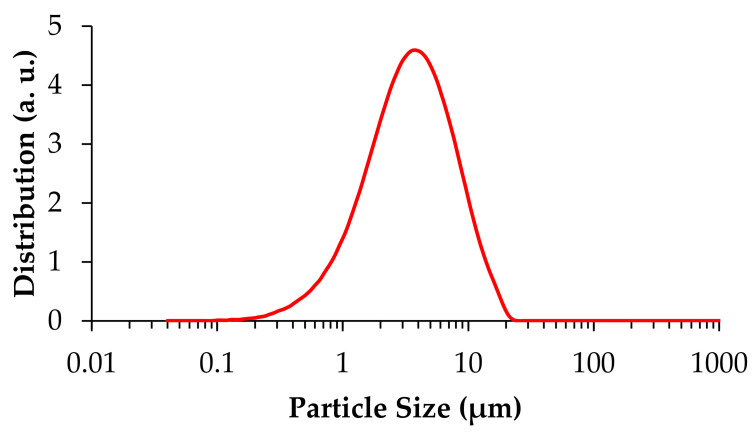
The particle size distribution of the milled powder.

**Figure 2 materials-14-05946-f002:**
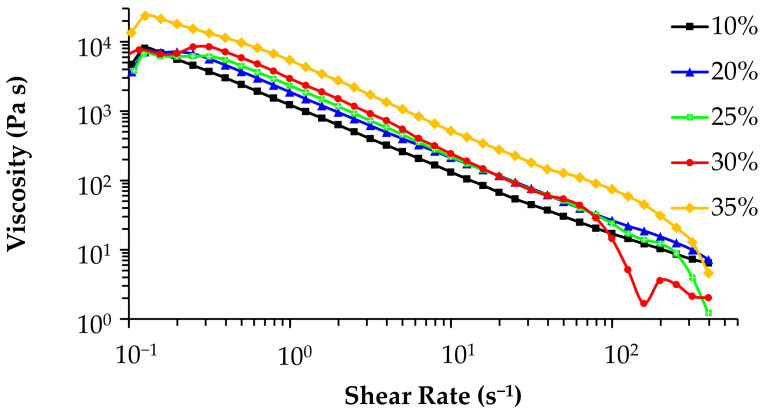
Dependence of the flow properties (viscosity vs. shear rate curves) on solid concentration.

**Figure 3 materials-14-05946-f003:**
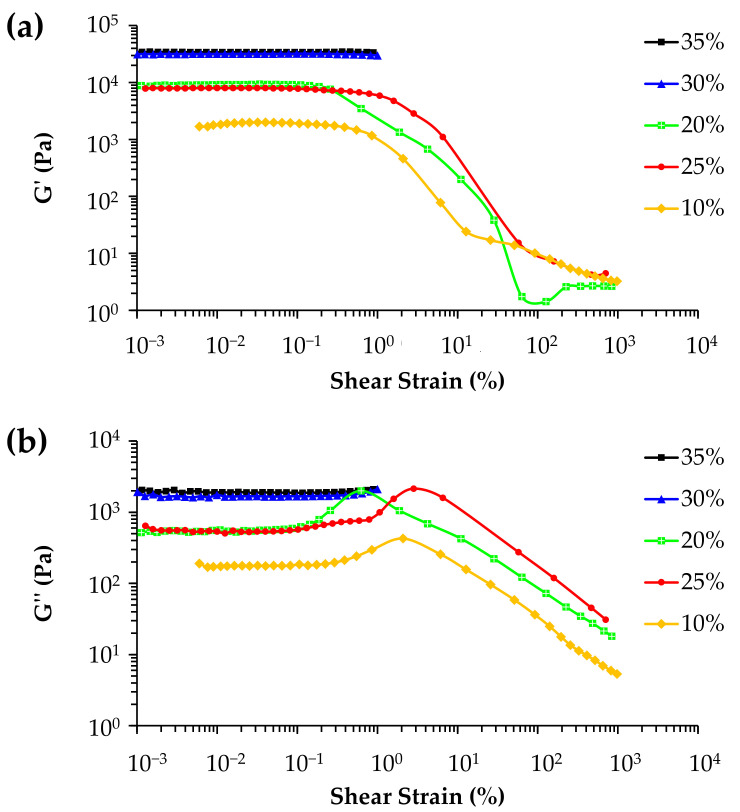
Shear strain dependence of the viscoelastic parameters: (**a**) elastic modulus, G′, and (**b**) storage modulus G″ on solid concentration.

**Figure 4 materials-14-05946-f004:**
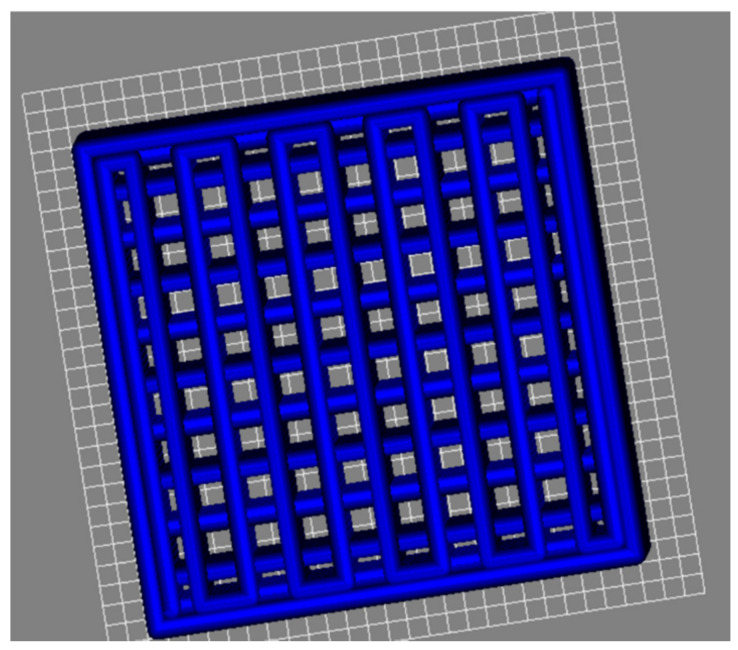
Planned scaffold structure designed according to the CAD model presented in Figure 1 of the article [26] using rods with *d* = 0.410 mm, a center to center distance between successive filaments, *s* = 0.820 mm, and an interlayer distance *h* = 0.527 mm, corresponding to ΔZ = 0.322 mm.

**Figure 5 materials-14-05946-f005:**
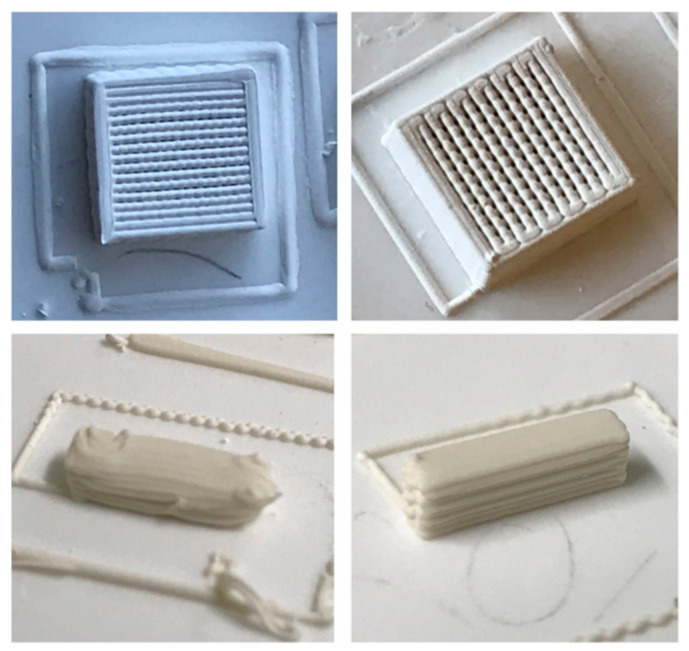
Effects of the overlap degrees set in the CAD model on geometrical definition of printed parts.

**Figure 6 materials-14-05946-f006:**
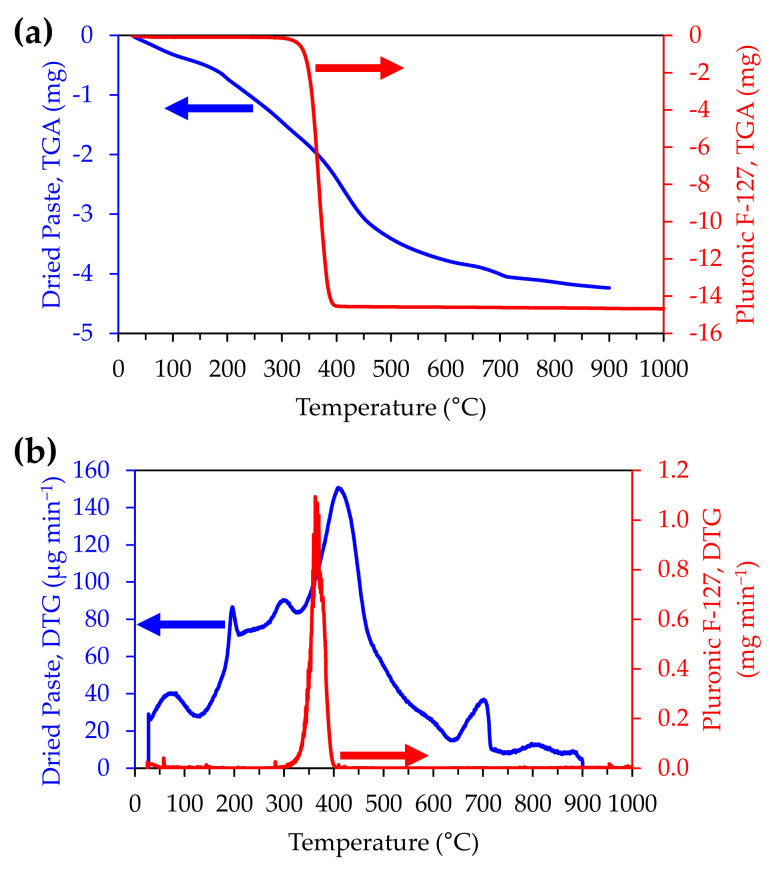
Comparative thermogravimetric analysis of pure Pluronic F-127 and of dried bioactive glass printing paste: (**a**) TGA, (**b**) DTG.

**Figure 7 materials-14-05946-f007:**
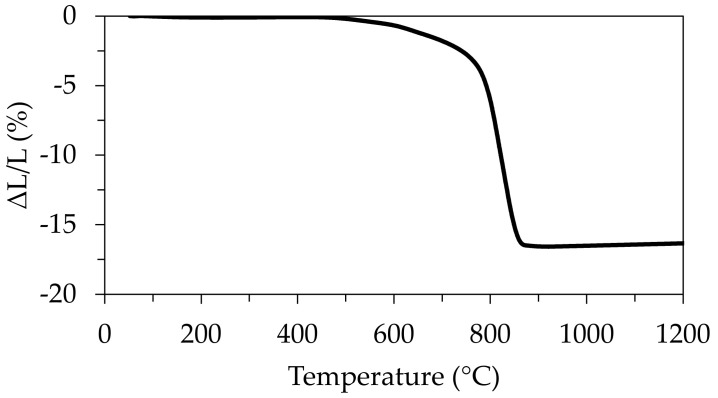
Dilatometric behaviour of the green sample printed from the paste containing 35 vol.% bioactive glass.

**Figure 8 materials-14-05946-f008:**
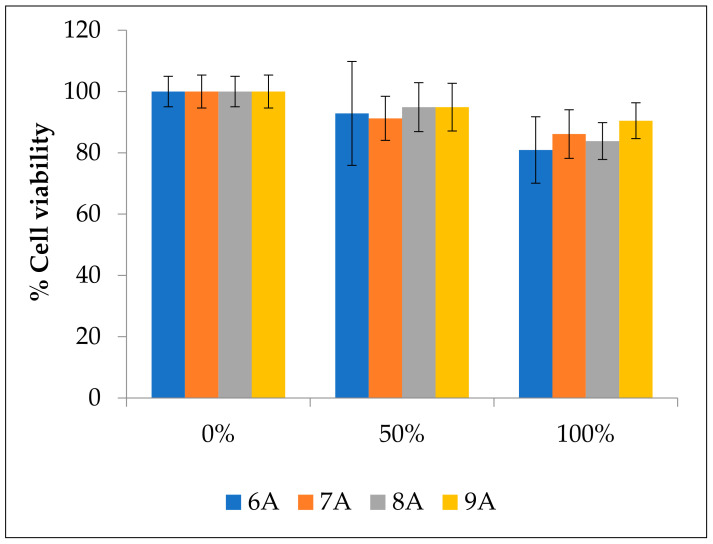
Cell viability values for scaffolds sintered at 600 (6A), 700 (7A), 800 (8A), and 900°C (900A) at different extract concentrations: 0% (0% extract, 100% DMEM medium), 50% (50% extract, 50% DMEM medium), 100% (100% extract, 0% DMEM medium).

**Table 1 materials-14-05946-t001:** Effects of wet milling liquids on milling efficiency.

Bioactive Glass	Wet Milling Liquids	Milling Conditions	Recovered Powder	Sieve Mesh	Powder Fraction > 63 µm
15 g	15 g ethanol	1 h 390 RPM	12.88 g (86%)	63 µm	6.14 g (~47%)
	15 g acetone		13.35 g (89%)		12.88 g (~97%)

**Table 2 materials-14-05946-t002:** Average density of the bioactive glass calcined at 600 °C.

Weight (g)	Average Volume (cm^3^)	Density (g/cm^3^)
1.59	0.595	2.672 ± 0.010

**Table 3 materials-14-05946-t003:** Dimensions and compressive strength data for bioactive glass scaffolds sintered at 800 °C.

Sample No.	Dimensions (mm)			Force (N)	Compressive Strength (MPa)
	x	y	z		
1	9.64	804	3.43	804	8.48
2	9.66	944	3.41	944	9.96
3	9.60	657	3.48	657	7.14
4	9.78	846	3.34	846	8.76
5	9.64	716	3.42	716	7.62
**Average Compressive Strength (MPa):**					**8.39 ± 0.20**

## Data Availability

Data is contained within the article. Supplementary data to this study are available in [doi:10.3390/ma14164515].

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
