# Peer review of "3D Printing of Macro Porous Sol-Gel Derived Bioactive Glass Scaffolds and Assessment of Biological Response"

_materials, 2021, doi:10.3390/ma14205946_

Round 1

Reviewer 1 Report

This study developed the printable bio-glass scaffold. The printing ink was optimized in the solid load concentration for the printing. The fabricated scaffold has sufficient mechanical strength and biocompatibility. I recommended to accept this paper for the Journal after a revision.

  1. Bio-glass

 The synthesized bio-glass was amorphous or crystalline? As well, after sintering of the printed scaffold, resultant bio-glass scaffold was amorphous?

  1. Number of samples

Please insert number of the samples in each evaluation.

  1. Porosity of the scaffold

The sintered scaffold had dense structure? The sintering of the bio-glass was completely accomplished by the heat-treatment?

  1. Shrinkage

 The shrinkage of the printed scaffold during heating process was indicated in Fig. 7. However, the shrinkage may be different in the direction, X, Y, Z directions. The dimensional shrinkage depends on the printed direction. Please discuss this.

  1. Cell viability

 Please discuss on the cell viability along with chemical composition of the bio-glass.

Author Response

Author's Reply to the Review Report (Reviewer 1)

This study developed the printable bio-glass scaffold. The printing ink was optimized in the solid load concentration for the printing. The fabricated scaffold has sufficient mechanical strength and biocompatibility. I recommended to accept this paper for the Journal after a revision. 

  1. Bio-glass

The synthesized bio-glass was amorphous or crystalline? As well, after sintering of the printed scaffold, resultant bio-glass scaffold was amorphous?

Response: Thanks for inquiring about this. Yes, the synthesized bioactive glass was completely amorphous as well demonstrated in our reference 25, and mentioned at the bottom of the section “2.1. Bioactive glass synthesis”. Crystalline phases only developed upon heat treating the sol-gel derived powder at 900 ºC, while sintering of the scaffolds was performed at 800 ºC.

  1. Number of samples

Please insert number of the samples in each evaluation.

 Response: As shown in Table 3, the number of scaffolds tested to assess the mechanical properties was 5, while in the biological characterization the number of scaffolds tested was 4, as mentioned in the manuscript. For the assessment of powder characteristics, triplicate measurements were performed as mentioned in the section “2.3. Particle characterization”.

  1. Porosity of the scaffold

The sintered scaffold had dense structure? The sintering of the bio-glass was completely accomplished by the heat-treatment?

Response: As mentioned in the manuscript, a scaffold model with a macro porosity fraction of about 70% was planned. The full derivation of the model is presented in the Appendix of the manuscript. Moreover, an approximate value of the porosity can be measured by measuring the external dimensions of the scaffold and its weight. However, because of the outside shell, it does not provide the accurate internal porosity. In other words, the deposited filaments are not completely full dense, but the overall porosity values are usually close to the analytical model.

  1. Shrinkage

The shrinkage of the printed scaffold during heating process was indicated in Fig. 7. However, the shrinkage may be different in the direction, X, Y, Z directions. The dimensional shrinkage depends on the printed direction. Please discuss this.

Response: Thanks for inquiring about this. Our preliminary dimensional assessments of the printed scaffolds before and after sintering revealed that anisotropic shrinkage was not an issue here.

  1. Cell viability

 Please discuss on the cell viability along with chemical composition of the bio-glass.

Response: In our manuscript, the cell viability was discussed along with the sintering temperature of the different scaffolds, as the chemical composition of the bioactive glass is the same for the different scaffolds. Our findings regarding cell viability agree with other previous studies with bioactive glass structures with a composition similar to those presented in our manuscript (see Ben-Arfa et al. (2017), ref. 23 in the present manuscript and Schmitz et al. (2020) (Bioactive Materials 5:55-65)). In our work, the ion release profile was not studied, but in the study conducted by Schmitz et al. (2020) it has been shown that the presence of Mg in the bioactive glass and consequent release to the culture medium contributed to the increase of cell viability.  

Reviewer 2 Report

The manuscript describes a new sol-gel bioactive glass bioink that may be suitable for bone repair applications. The work and manuscript are both high quality, and detailed material characterization is presented. There are a few concerns which must be addressed prior to further consideration.

  1. its a bit troubling about the lack of accuracy of the 3D printer in the y direction which leads to variation in materials extruded and also mechanical properties. this should be at least discussed.
  2. placement of printed/sintered samples in SBF is a good start, followed by placing mixtures of this extract for cell compatibility. However, what the the mineralization profile (calcium and phosphate levels) on the scaffolds following SBF incubation? this could tell alot about the potential of the scaffolds as a bone substitute. 
  3. Its not clear how porosity of 70% was determined?
  4. for cell assays, it would be most appropriate to seed the osteoblastic cells onto the scaffolds and measure viablity (Live/dead) and gene expression to ensure an appropriate cell/material interaction.
  5. in the abstract, near the end, it states human keratinocytes, which were not used in this study.

Author Response

Author's Reply to the Review Report (Reviewer 2)

The manuscript describes a new sol-gel bioactive glass bioink that may be suitable for bone repair applications. The work and manuscript are both high quality, and detailed material characterization is presented. There are a few concerns which must be addressed prior to further consideration.

  1. its a bit troubling about the lack of accuracy of the 3D printer in the y direction which leads to variation in materials extruded and also mechanical properties. this should be at least discussed.

Response: We tend to partially agree with the reviewer in that 3D printer has some conceptual limitations in the y direction concerning shape-free modeling of scaffolds. The authors were very aware and comfortable about these intrinsic limitations before starting the study. Please note that the key point to take full profit from Robocasting and obtain good mechanical properties of 3D-printed scaffolds with similar levels of porosity is to prepare inks with suitable viscoelastic properties, as demonstrated recently in our references 15, 29 and 38. These aspects have been better highlighted in the discussion part.

  1. placement of printed/sintered samples in SBF is a good start, followed by placing mixtures of this extract for cell compatibility. However, what the the mineralization profile (calcium and phosphate levels) on the scaffolds following SBF incubation? this could tell alot about the potential of the scaffolds as a bone substitute.

Response: The authors do agree in that measuring the ionic concentrations of calcium and phosphorous at different time points along the immersion period in SBF is a complementary way to document the evolution of the mineralization process, as we have shown in our reference 23. However, such information is not so essential, and there are many published articles that do not report such data. In the present case, the ICP equipment was broken and the experiments could not be performed. But this is not a concern at all. Please note that in comparison to reference 23, in the present work we have synthesized a more well-balanced alkali-free bioactive glass composition.

  1. Its not clear how porosity of 70% was determined?

Response: The internal porosity was analytical calculated using the scaffold model. The full derivation of the model is presented in the Appendix of the manuscript. Moreover, an approximate value of the porosity can be measured by measuring the external dimensions of the scaffold and its weight. However, because of the outside shell, it does not provide the accurate internal porosity. These values are usually close to the analytical model.

  1. for cell assays, it would be most appropriate to seed the osteoblastic cells onto the scaffolds and measure viablity (Live/dead) and gene expression to ensure an appropriate cell/material interaction.

Response:  We are thankful for the reviewer’s suggestion. The aim of this experiment was to confirm the biocompatibility of the bioactive glass scaffolds and for that, the MTT test was performed by exposing the cells to the medium extract obtained by incubating the different scaffolds for 24 h. This approach assumes that ions may be released by the scaffolds to the culture medium and eventually interfere with cell viability. This indirect approach has been followed by several authors for the study of the biocompatibility of bioactive glasses, as for instance Ciraldo et al. 2018 (Acta Biomaterialia 75:3-10) and Schmitz et al. 2020 (Bioactive Materials 5:55-65). As our results show that for all scaffolds, the cell viability values are above the 70 % threshold which, according to ISO 10993-5:2009(E) (2009) suggests they are safe and thus suitable for biomedical applications, we have considered that the MTT assay responded to our aim. Nevertheless, seeding the cells in the scaffolds is also an interesting experiment to understand the cell-material interactions and we will consider it in our future experiments.

  1. in the abstract, near the end, it states human keratinocytes, which were not used in this study.

Response: Thank you for the warning, it has been corrected.